# Expression of Recombinant Clostridial Neurotoxin by *C. tetani*

**DOI:** 10.3390/microorganisms12122611

**Published:** 2024-12-17

**Authors:** Brieana M. Gregg, Sonal Gupta, William H. Tepp, Sabine Pellett

**Affiliations:** 1Department of Bacteriology, University of Wisconsin-Madison, Madison, WI 53706, USA; 2Wake Forest Institute for Regenerative Medicine, Winston Salem, NC 27157, USA

**Keywords:** botulinum neurotoxin, recombinant, tetanus toxin, *Clostridium tetani*, protein stability, expression system

## Abstract

Tetanus neurotoxins (TeNT) and botulinum neurotoxins (BoNTs) are closely related ~150 kDa protein toxins that together comprise the group of clostridial neurotoxins (CNTs) expressed by various species of *Clostridia*. While TeNT is expressed as a single polypeptide, BoNTs are always produced alongside multiple non-toxic proteins that form a stabilizing complex with BoNT and are encoded in a conserved toxin gene cluster. It is unknown how *tent* evolved without a similar gene cluster and why complex-free TeNT is secreted as a stable and soluble protein by *C. tetani*, whereas complexing proteins appear to be essential for BoNT stability in culture supernatants of *C. botulinum*. To assess whether the stability of TeNT is due to an innate property of the toxin or is a result of *C. tetani*’s intra- and extra-cellular environment, both TeNT and complex-free BoNT/A1^ERY^ were expressed recombinantly in atoxic *C. tetani* and analyzed for expression and stability. The strong clostridial ferredoxin (*fdx*) promotor resulted in the expression of recombinant TeNT at greater levels and earlier time points than endogenously produced TeNT. Recombinant BoNT/A1^ERY^ was similarly expressed by atoxic *C. tetani*, although partial degradation was observed. The rBoNT/A1^ERY^ produced in *C. tetani* was also partially proteolytically processed to the dichain form. Investigations of bacterial growth media and pH conditions found that the stability of rTeNT and rBoNT/A1^ERY^ in spent media of *C. tetani* or *C. botulinum* was affected by growth media but not by pH. These data indicate that the distinct metabolism of *C. tetani* or *C. botulinum* under various growth conditions is a primary factor in creating a more or less favorable environment for complex-free CNT stability.

## 1. Introduction

Tetanospasmin, or tetanus neurotoxin (TeNT), is a bacterial toxin expressed by the anaerobic, Gram-positive, bacilli *Clostridium tetani* (*C. tetani*). TeNT and the closely related botulinum neurotoxins (BoNTs), together known as the clostridial neurotoxin (CNT) family, are the most potent natural toxins known to humans [1,2]. TeNT is the agent responsible for tetanus, which is a potentially deadly disease causing long-lasting spastic paralysis of skeletal muscles. The dramatic clinical presentations of tetanus, including locked jaw and extreme muscle spasms, have been noted in medical manuscripts as early as the third to fifth century BC. However, it was not until the late 19th century that the cause of the disease was determined to be the TeNT produced by *C. tetani* [3,4]. *C. tetani* spores are found nearly ubiquitously across the world in soil, dirt, and manure, and they may germinate and begin TeNT expression when exposed to sufficiently anaerobic environments such as a deep wound [4,5].

TeNT is expressed by vegetative *C. tetani* cells as a ~150 kDa single-chain polypeptide, which the bacteria proteolytically process into a 50 kDa light chain (LC) and 100 kDa heavy chain (HC) linked by a disulfide bond [6]. If humans are exposed to the toxin through the infection of a deep wound, TeNT holotoxin is released into the circulation and makes its way to the peripheral nervous system, where it is endocytosed into motor-neuron nerve endings, which is followed by intra-axonal retrograde transport and transcytosis to inhibitory interneurons of the central nervous system [7,8,9]. Here, the zinc endopeptidase LC is released into the cytosol of inhibitory neurons and cleaves Vesicle-Associated Membrane Protein 2 (VAMP2). VAMP2 is a critical component in assembling Soluble NSF Attachment protein Receptor (SNARE) complexes [10], and its cleavage halts the SNARE-mediated membrane fusion of neurotransmitter-containing vesicles, thus blocking the release of inhibitory neurotransmitters gamma-aminobutyric acid (GABA) and glycine [11,12]. This results in continued neuromuscular excitation and the spastic paralysis characteristic of tetanus. TeNT is extremely potent, with an estimated minimum mouse lethal dose of only 0.2 ng/kg in humans [13], although the development of a successful toxoid vaccine has markedly lowered adult and neonatal tetanus case numbers in countries with robust vaccination programs [5].

BoNTs are a family of closely related protein toxins and are the agents responsible for botulism, which is a rare but extremely serious disease characterized by descending flaccid paralysis [1,2,14]. BoNTs are produced by several species of *Clostridium*, including *Clostridium botulinum* (*C. botulinum*) and *Clostridium baratii* (*C. baratii*), and have been traditionally categorized into seven immunogenically distinct serotypes (BoNT/A-G) containing over 45 subtype variations and mosaic toxins [15,16]. Like TeNT, BoNTs are produced as ~150 kDa single-chain proteins that are subsequently converted by endogenous or exogenous proteases to the more active AB-type dichain toxins consisting of a catalytically active LC linked via disulfide bond to an HC responsible for neuronal cell binding and LC translocation [17,18]. BoNTs similarly target vertebrate neurons, although intoxication occurs primarily at cholinergic nerve terminals, where the LC-mediated cleavage of various SNARE targets prevents the fusion of acetylcholine-containing vesicles to the membrane of the presynaptic nerve terminal [19,20]. The absence of acetylcholine signaling in the synaptic cleft results in flaccid muscular paralysis as opposed to the spastic paralysis brought about by TeNT intoxication.

Although TeNT and BoNTs share a great deal of similarity regarding overall toxin structure and function, there are several key differences. BoNTs are highly diverse, and the recent use of large bioinformatic databases has led to the discovery of *bont* genes across the plasmids, bacteriophages, and chromosomes of multiple *Clostridial* species [21,22,23,24,25,26]. Across serotypes, *bont* toxin genes range in nucleotide identity from ~32 to 75% [27]. All *bont* serotypes and subtypes are found clustered with genes coding for a variety of neurotoxin-associated proteins (NAPs), which stabilize and protect BoNTs from chemical and enzymatic threats [1,21] and aid BoNTs passage through gastrointestinal tracts after oral exposure [22]. NAPs encoded within the gene cluster vary by toxin serotype [23], although all presently discovered serotypes possess a gene encoding for non-toxic non-hemagglutinin (NTNH) protein. NTNH is a ~130–150 kDa protein sharing much of its domain structure with BoNTs, and it binds the associated BoNT to form a ~300 kDa minimal progenitor complex (M-PTC) [28,29]. The binding of additional NAPs to NTNH results in a full progenitor complex (PTC) up to ~900 kDa in size [28].

In contrast, the *tent* operon is simpler, consisting only of a plasmid-encoded promoter region, the positive regulator *tetR*, and the adjacent *tent* gene [30,31]. Unlike *bont* genes, *tent* is highly conserved among toxigenic *C. tetani* strains, and no immunogenically distinct TeNT serotypes have been reported [32]. There also is no evidence of TeNT forming a complex with NAPs analogous to those of BoNTs or otherwise. This prompts the question of what allows the stable production of TeNT without the need for complexing proteins, when there appears to be universal evolutionary pressure for the retention of BoNT NAP genes [22]. The goal of this study was to determine if the stability of TeNT in *C. tetani* culture supernatants is due to the environment created by *C. tetani* metabolism or to the resistance of TeNT to proteases that may be present in a *C. tetani* culture supernatant.

Expression analyses of rCNTs in an atoxic *C. tetani* strain indicate high production levels of both rTeNT and rBoNT

^ERY^ dependent on the promoter used and with at least partial proteolytic processing to the dichain. Comparative analysis of the relative stability of BoNT/A1 and TeNT in spent *C. botulinum* and *C. tetani* culture media suggested that both rCNTs were susceptible to degradation and that modified Mueller Miller media enhanced the stability of rCNTs. Our data reveal that the stability of complex-free TeNT and BoNT/A was similarly affected by the environment created as a result of metabolism during the fermentation of *C. tetani* and *C. botulinum*, suggesting that an atoxic *C. tetani* strain lacking the endogenous *tent* may be useful as a high-level production system of complex-free rBoNT and rTeNT.

## 2. Materials and Methods

### 2.1. Biosafety and Security

Our Wisconsin laboratory and personnel are registered with the CDC Select Agent Program for research involving botulinum neurotoxins and botulinum neurotoxin-producing strains of clostridia. The research program, procedures, occupational health plan, documentation, security, and facilities are closely monitored by the University of Wisconsin—Madison Biosecurity Task Force, University of Wisconsin—Madison Office of Biological Safety, the University of Wisconsin Select Agent Program, and at regular intervals by the CDC and the Animal and Plant Health Inspection Service (APHIS) as part of the University of Wisconsin—Madison Select Agent Program. All personnel have undergone suitability assessments and completed rigorous and continuing biosafety training, including biosafety level 3 (BSL3) and select agent practices before participating in laboratory studies involving botulinum neurotoxins and neurotoxigenic *C. botulinum* strains. All recombinant DNA protocols for the construction of the recombinant BoNT genes and their expression in *C. botulinum* strains have been approved by the University of Wisconsin Institutional Biosafety Committee (IBC), and specific experiments were approved by the Division of Select Agents and Toxins at the CDC. A dual use research of concern (DURC) risk mitigation plan has been established and approved by the University of Wisconsin—Madison Select Agent Program and NIAID for these experiments.

### 2.2. Media and Reagents

Culture media reagents were purchased from Sigma Aldrich (St. Louis, MO, USA) and BD Difco (Franklin Lakes, NJ, USA). HySoy (PHBT30) and HySoyT (PHBT31) were purchased from Kerry (Formerly QuestSheffield; Tralee, Ireland). Antibiotics (chloramphenicol, cycloserine, kanamycin, thiamphenicol) were purchased from Sigma Aldrich (St. Louis, MO, USA) and used at the following concentrations: in *E. coli*: kanamycin 30 µg/mL, chloramphenicol 25 µg/mL in agar and 12.5 µg/mL in liquid media; *C. botulinum*: thiamphenicol 15 µg/mL; trimethoprim (1.5 µg/mL), sulfamethoxazole (30 µg/mL), and cycloserine (250 µg/mL).

### 2.3. Bacterial Strains

ATCC *C. tetani* 454: BioProject PRJNA158341, Gen Bank assembly GCA_000987095.1; ATCC *C. tetani* 19406: BioProject PRJNA260197, GenBank assembly GCA_000762305.1; *C. botulinum* LNT01 [33] is an atoxic strain, lacking the entire *bont* gene cluster including both the toxin and all associated complexing proteins. *E. coli* conjugative donor strains were purchased from Plasmid Vectors (University of Nottingham, UK). Briefly, these are derivatives of commercial strains wherein the conjugative plasmid R702 has been inserted to aid in transfer of the plasmid to Clostridial species. The resulting strains, along with the commercial strain they were derived from, are as follows: *E. coli* CA434 (HB101), S. Express (NEB, Express). Further information regarding these strains, including genotypes, may be found at https://plasmidvectors.com/donor-strains/ (accessed on 16 December 2024). *E. coli 10* beta was purchased from New England Biolabs (Ipswich, MA, USA). *E. coli* was grown at 37 °C in either liquid LB agitated at 225 rpm or on LB agar plates supplemented with corresponding antibiotics for plasmid selection. Clostridia cultures were grown in static nitrogen-flushed hungate tubes, and the handling of culture was carried out within an anaerobic chamber (Forma Anerobic System, Marietta, OH, USA). The atmosphere consists of 80% N_2_, 10% CO_2_, and 10% H_2_. *E. coli*, *C. botulinum*, and *C. tetani* cultures were grown in the indicated liquid media or agar plate and formulated as follows. Tryptone peptone glucose yeast media (TPGY): 5% trypticase peptone, 0.5% bacto peptone, 0.4% glucose, 2% yeast extract, 0.1% L-cysteine HCl (pH 7.3–7.4); toxin production media (TPM): 2% trypticase peptone, 0.5% glucose, 1% yeast extract (pH 7.3); tryptone, yeast, glucose media (TYG): 3% bacto tryptone, 2% yeast extract, 1% glucose, 0.1% sodium thioglycolate; HySoy: 5% HySoy (Kerry, Tralee, Ireland), 0.75% glucose, 0.0125% L-cysteine, 0.0125% L-tyrosine, 0.006% FeSO_4_, 0.5% NaCl, 0.05% Na_2_HPO_4_, 0.0175% KH_2_PO_4_, 0.05% MgSO_4_ 7H_2_O (pH 6.8). Modified Mueller Miller media was prepared as described in Lantham et al. [34]: (per liter) NZ Casein TT 43.5 g, glucose 9.7 g, NaCl 2.5 g, reduced iron powder 0.5 g, MgSO_4_ 0.1g, L-cysteine 125 mg, uracil 1.25 mg, calcium pantotenate 1 mg, vitamin B12 0.05 mg, nicotinic acid 0.25 mg, riboflavin 0.25 mg, thiamine HCl 0.25 mg, pyridoxin HCl 0.25 mg, biotin 2.5 µg. When noted, media were prepared with reduced iron powder at a concentration of 0.5 g/L, which replaced the use of FeSO_4_. We used 0.5 mg/mL resazurin as an anerobic indicator in agar plates.

### 2.4. Antibiotic Sensitivity of C. tetani and E. coli Strains

Triplicate samples of each strain were grown on anaerobic TPGY (*C. tetani*) or aerobic LB (*E. coli*) agar plates at 37 °C for 72 h. Strains were considered weakly resistant if there was minimal bacterial growth compared to resistant strains or if growth did not appear within 24–48 h after plating.

### 2.5. Generation of TeNT Plasmid Constructs

Isolation of genomic DNA (gDNA) from wild type *C. tetani* 64001: Unless otherwise stated, all centrifugation steps in gDNA isolation were performed at 16,000× *g. C. tetani* 64001 was grown for 7 h in TPGY liquid media; then, 1 mL of aliquot was centrifuged for 5 min, and the cell pellet was resuspended in 300 µL 1X TES buffer (25 mM Tris-HCl, pH 8.0, 0.1 M NaCl, and 0.001 M EDTA), 10 µ L RNase A (100 mg/mL, Qiagen, Germantown, MD, USA) and 100 µL lysozyme (20 mg/mL, Sigma Aldrich, St. Louis, MO, USA). Following 30-minute incubation at 37 °C, samples were treated with 50 µL 10% sodium dodecyl sulfate (SDS, ThermoFisher, Waltham, MA, USA) and 50 µL proteinase K (10 mg/mL, Qiagen, Germantown, MD, USA), which was followed by another 30-min incubation at 25 °C. Then, 100 µL 5M sodium perchlorate was added, and the sample was incubated for 10 min at 25 °C. Afterwards, 600 µL phenol-chloroform-isoamyl (25:24:1) mixture was dispensed into a Phasemaker tube (Invitrogen Thermo Fisher,#A33248, Waltham, MA, USA) along with 600 µL of the culture lysate sample and hand inverted for 5 min before 5 min centrifugation. The supernatant was carefully transferred into 600 µL fresh phenol-chloroform-isoamyl (25:24:1) before inversion, and then the centrifugation steps were repeated. The resulting supernatant was transferred into pre-chilled 100% EtOH and inverted until gDNA precipitation was visible and could be removed for overnight incubation in 4 °C 70% EtOH. The sample was briefly centrifuged, EtOH was aspirated off, and any residual EtOH was allowed to evaporate before resuspension of the gDNA in 10 mM Tris buffer, pH 8.0.

Primers ‘NdeI-tent’ and ‘tent-NheI’ were purchased from Integrated DNA Technologies Inc. (Coralville, IA, USA) and used to amplify the tent gene from the gDNA. They include the addition of a restriction sites NdeI directly upstream and NheI directly downstream of the tent to aid in the insertion of the gene fragment into cloning vectors.

NdeI-tent (5′CGCATATGATACGTATGCCAATAACCATAAATAATTTTAGATATAGTG3′)

tent-NheI (5′CGGCTAGCTTAATCATTTGTCCATCCTTCATCTGTAGG3′)

PCR amplification was performed using a Q5 High Fidelity PCR master mix (NEB), 10 µM primers and ~100 ng *C. tetani* 64001 gDNA in a GeneAmp PCR 9700 (Applied Biosystems, Foster City, CA, USA). The resulting ~4 kb PCR product was eluted from agarose gel using the Monarch DNA Gel Extraction Kit (NEB) and confirmed via Sanger sequencing. Then, tent was digested with restriction enzymes NdeI and NheI to facilitate insertion into the PJET1.2 (ThermoFisher, Waltham, MA, USA) cloning vector containing either the thl or fdx promotor. All cloning steps were carried out using the CloneJet PCR Cloning Kit (K1231, ThermoFisher, Waltham, MA, USA) according to the manufacturer’s instructions. The resulting PJET vectors were digested with SbfI and NheI to excise thl-tent or fdx-tent regions, which were inserted into pMTL82151. This generated pMTL82152-TeNT and pMTL82153-TeNT. tetR is located slightly upstream of tent on the E88 plasmid and is preceded by its own promotor (ptetR) and followed by the tent promotor (ptent) [30,35]. We synthesized a segment ~200 bp prior to tetR to the start of tent, which was flanked by Sbf1 and NdeI restriction enzyme sites for seamless insertion of the region into the shuttle vector. tetR contains an internal NdeI restriction site (CATATG), so a silent point mutation was introduced to eliminate the site (CACATG). The ~1 kb region (referred to as tetR) was synthesized by GeneArt (ThermoFisher, Waltham, MA, USA), using the publicly available *C. tetani E88* plasmid pE88 sequence (GenBank Accession AF528097.1). tetR was cloned directly into pMTL82151 using SbfI and NdeI restriction enzymes, and subsequent cloning inserted tent using NdeI and NheI restriction enzymes. This generated pMTL82155-TeNT.

### 2.6. Generation of BoNT/A1^ERY^ Plasmid Constructs

The gene for catalytically inactive BoNT/A1 (*bont/a1*^ERY^) possesses three point mutations at E224Q, R363A, and Y366F. This toxin has not been shown to induce detectable toxicity [21,36] and thus is excluded from Tier 1 Select Agent regulations regarding the expression of rBoNTs. The *bont/a1*^ERY^ was excised out of PJET 1.2-*bont/a1*^ERY^via restriction enzymes NheI and NdeI. This digest excised *bont/a1*^ERY^, which was inserted into pMTL vectors already containing the fdx promotor region. This generated pMTL82153-BoNT/A1^ERY^. All constructs were confirmed via whole plasmid sequencing.

### 2.7. Conjugation of pMTL Plasmids into C. tetani

Plasmids were transformed into *E. coli* donor strains CA434 (for BoNT/A1^ERY^ constructs) [37] or S. Express (for TeNT constructs) as previously described in LB media supplemented with 30 µg/mL kanamycin (maintenance of the R702 conjugative plasmid) and 15 µg/mL chloramphenicol (analogous to thiamphenicol; used in *E. coli* to maintain pMTL vectors). Plasmid constructs which displayed increased instability were prescreened prior to mating via plasmid isolation (QIA Spin MiniPrep Kit, Qiagen, Germantown, MD, USA) followed by restriction enzyme digest. Plasmid conjugation was performed essentially as described in previous studies [37,38] with the following alterations. *C. tetani* 454 and 19406 were anaerobically cultured at 37 °C in unsupplemented tryptone peptone glucose yeast (TPGY) media to the mid-log phase of OD600 0.6 to 0.7 as an overgrowth of the recipient strain, which resulted in a decrease in the overall conjugation efficiency. Liquid cultures of the transformed *E. coli* donor strains were then anaerobically co-incubated with liquid culture of either *C. tetani* 454 or 19406 on unsupplemented TPGY agar plates. Samples were allowed to mate for 24 h at 37 °C before cultures were resuspended and transferred to fresh TPGY plates (3% agar) supplemented with the TSC + thiamphenicol antibiotic combination. After growth was visible on the plates, random colonies were selected and cultured in TPGY, and then their plasmids were sequenced to confirm maintenance of the toxin gene.

### 2.8. Sequencing

Whole plasmid sequencing was performed using the Oxford Nanopore MinION and Rapid Barcoding Kit 24 (SQK-RBK114.24, Oxford Nanopore, Oxford, UK) according to the manufacturer’s instructions. Read assemblies were performed in EPI2ME, and sequences were analyzed in UGENE V47.0 and MacVector software. Sanger sequencing was performed by the University of Wisconsin-Madison Biotechnology Center.

### 2.9. SDS-PAGE and Western Blot

Purified inactive 8MTT TeNT [39] was used as the TeNT toxin control, and purified BoNT/A1 holotoxin [40] was used as the BoNT toxin control. After culturing, the whole culture, pellet, and supernatant samples were collected at the indicated time points. Whole cultures were centrifuged at 16,000× *g* for 10 min in order to separate the cell pellet and supernatant. Cell pellets were resuspended in an equal volume of 1X TES buffer with 2 mg/mL lysozyme (Sigma Aldrich, St. Louis, MO, USA) and incubated at 35 °C for 10 min. For the collection of cell growth from solid agar plates, each plate was flooded with 10 mL 1X PBS and manually resuspended with a plate scraper. This whole culture was then processed into supernatant and pellet samples as above. Expression samples were mixed with 4X NuPAGE LDS sample buffer (Invitrogen Thermo Fisher, Waltham, MA, USA) to 1X with or without the addition of 100 mM dithiothreitol (DTT) as a reducing agent. Samples were heated to 95 °C for 5 min and run on 4–12% Bis-Tris (Invitrogen) gels before transfer to 0.45 µm PVDF membrane. Western blots were probed with primary mouse polyclonal anti-8MTT (prepared in the lab) or rabbit polyclonal anti-BoNT/A1 antibody (prepared in the lab) at 1:1000 dilution. Secondary goat anti-mouse, and bovine anti-rabbit alkaline phosphatase antibodies (Santa Cruz Biotechnology, Dallas, TX, USA) were used at 1:7500 dilutions. Blots were exposed to KPL PhosphoGlo AP substrate (SeraCare Life Sciences, Milford, MA, USA) prior to imaging.

### 2.10. Toxin Stability in Spent Culture Media

Toxin stability experiments were conducted similar to previous [41]. Briefly, 10 mL nitrogen-flushed hungate tubes of the specified media were inoculated with 100 µL of either *C. tetani* 454 (82153) or *C. botulinum* LNT01 [33]. LNT01 is an atoxic strain, lacking the *bont* and associated complexing genes. Replicates (n = 3) were cultured statically at 37 °C for 96 h, after which the whole culture was centrifuged at 6000× *g* for 10 min, and culture supernatants were pH adjusted to pH 5, 6, 7, and 8 via 10 N NaOH and 1 N HCl. Each replicate group additionally included a spent media sample which was not pH adjusted and one sample of fresh media which had never supported bacterial growth. All samples were sterile filtered through 0.22 µm polyvinylidene difluoride (PVDF) filters. Then, 50 ng (for BoNT/A1) or 250 ng (for TeNT-8MTT) of sterile, purified toxin were added to each sample and left to incubate at 37 °C for 24 h. Samples were analyzed via SDS-PAGE and Western blotting, and the concentration of the ~150 kDa protein bands was quantified via densitometry (Azure Spot v 2.0.062).

## 3. Results

### 3.1. C. tetani 454 and 19406 Are Compatible Recipient Strains for Conjugative Plasmid Transfer

Two atoxic *C. tetani* strains were chosen for our analysis of rTeNT and rBoNT expression due to their unique properties and relatedness to the Harvard strain, which is used industrially to produce TeNT [42,43]. Strain ATCC 19406 is a North American laboratory diagnostic strain closely related to the ‘Harvard’ strain but lacking the entire 74 kb toxin-encoding plasmid [44]. *C. tetani* ATCC 454 is an atoxic strain isolated from Chinese clinical fecal samples in the 1920s and also closely related to the Harvard strain, although it only lacks an approximately 20 kb segment of the plasmid including both *tetR* and *tent* [44,45].

As little information is published regarding the antibiotic susceptibility of *C. tetani* strains outside of a clinical context, we first screened both strains of *C. tetani* as well as potential *E. coli* conjugative donor strains CA434, S. Express, Interstellar, and TopSex [37,46] for susceptibility to a variety of commonly used antibiotics and antibiotic combinations at various concentrations (Appendix A). Both *C. tetani* and *E. coli* are sensitive to thiamphenicol at 15 μg/mL, so we opted to use plasmid vectors containing the thiamphenicol resistance gene *catP* to select for maintenance of the desired plasmid [47]. Additionally, we chose the following antibiotic combination to select against the *E. coli* donor strain as it was effective to select against all tested donor strains: trimethoprim (1.5 µg/mL), sulfamethoxazole (30 µg/mL), and cycloserine (250 µg/mL) (TSC). Utilization of TSC + thiamphenicol allowed for the selection of only *C. tetani* containing the desired plasmid after conjugation.

To determine which origins of replication may be utilized in these strains, we screened several modular pMTL8000 series clostridial shuttle vectors [47] by conjugating them into *C. tetani* 19406 and 454. The pMTL82151 (*C. botulinum* replicon pBP1) and pMTL87151 (*C. perfringens* replicon pIP404) [47,48] reliably yielded trans-conjugants, indicating their functional origins of replications in these strains. However, pMTL87151 contains an internal NdeI restriction site which would complicate cloning, and therefore pMTL82151 was selected as the shuttle vector for future experiments. Finally, three promotors were examined for expressing rTeNT. The pMTL82152 encodes the ribosome binding site (RBS) and thiolase promotor (*thl*) of *C. acetobutylicum* [49], while pMTL82153 encodes the RBS and ferredoxin promoter (*fdx*) of *C. pasteurianum* [50]. The pMTL82155 was created by cloning the region encoding the *C. tetani* positive regulator *tetR*, the endogenous *tetR* promotor (p*_tetR_*), and the *tent* promotor (p*_tent_*) from wild-type *C. tetani* strain (64001) into pMTL82151 [35].

Following successful cloning and sequence verification of the *tent* constructs, it was observed that several plasmids underwent rearrangement following transformation into the *E. coli* donor strains. *E. coli* CA434, a *recA +* strain, was used as the preferred donor strain and resulted in the successful conjugation of pMTL82153-TeNT into both *C. tetani* 454 and 19406. The pMTL82155-TeNT was conjugated into *C. tetani* 454 via *E. coli* S. Express. The pMTL82152-TeNT plasmid was consistently rearranged in any of the tested *E. coli* donor strains and thus was not further pursued.

### 3.2. The Fdx Promotor Stimulates rTeNT Expression Earlier and in Greater Quantities than tetR and Wild-Type Strains

The expression of rTeNT in *C. tetani* strain 454 and 19406 was examined in three different growth media using three different promoters. Mueller and Miller developed a media formulation for increased expression of TeNT [42], which has been analyzed and improved upon in the decades since its creation. Of specific interest were formulations by Latham et al. (‘modified Mueller Miller media’), which removed several inorganic salts and amino acids [34], and by Demain et al. (‘HySoy media’), which replaced animal and dairy products with soy-based alternatives [51,52]. A final option, ‘HySoy-T’, is a commercially available media which utilizes an alternative source of soy peptides, but it is otherwise identical to the HySoy formulation. Each of these media was supplemented with 0.5 g/L of reduced iron powder, which has been shown to increase toxin production in *C. tetani* [52,53].

Western blot analysis of whole bacterial culture revealed that the expression of ~150 kDa TeNT holotoxin was greater in *C. tetani* strains containing the pMTL82153-TeNT plasmid compared to the pMTL82155-TeNT plasmid or expression in wt *C. tetani* strains, although the toxins varied in quantity and stability depending on the culture formulation (Appendix A). The highest and most consistent expression levels of rTeNT were observed in modified Mueller Miller (+reduced iron), where rTeNT expression in *C. tetani* 454/19406 (82153-TeNT) was first observed at 24 h and remained constant until 96 h of culture (Figure 1). In contrast, endogenous TeNT production by a wild-type *C. tetani* and rTeNT production driven by the endogenous promoter (*C. tetani* 454 (82155-TeNT)) was not reliably detected in Mueller Miller media using the same Western blot conditions (Figure 1), but it was observed after 96 h of growth in HySoy or HySoy-T media (Appendix A). This toxin production coincided with a rapid pH drop from the roughly neutral pH up to 72 h to pH ~4.8–5.2 by 96 h. Even then, the quantity of TeNT expressed was lower than that of either 454 (82153-TeNT) or 19406 (82153-TeNT). These data indicate that rTeNT expression under the control of the *fdx* promotor in atoxic *C. tetani* results in a higher and more reliable production of the toxin independent of the metabolic shifts required for endogenous TeNT production [54,55]. Both *C. tetani* 454 and 19406 expressed rTeNT at roughly the same levels with no discernable difference in toxin break-down between the two. However, *C. tetani* strain 19406 had a tendency to swarm after conjugation, making it difficult to isolate single colonies. Thus, strain 454 was chosen for additional experiments.

Western blot of recombinant TeNT expression by *C. tetani* 454 and *C. tetani* 19406 was cultured in modified Mueller Miller (supplemented with 0.5 g/L reduced iron powder) media for 24–96 h. Expression construct pMTL 82153-TeNT contains the ferredoxin (*fdx*) promotor, while pMTL 82155-TeNT contains the *C. tetani* positive regulator *tetR* as well as the endogenous *tetR* promotor (p*_tetR_*) and the *tent* promotor (p*_tent_*) regions. To maintain the recombinant plasmids, *C. tetani* 454 and 19406 culture media were supplemented with 15 ug/mL thiamphenicol. Wild-type *C. tetani* WR23 was used as an expression control. All samples are unreduced, whole culture.

### 3.3. Full-Length Recombinant BoNT/A1^ERY^ Is Produced by C. tetani 454 Without Complexing Proteins

To examine if other CNTs may be stably expressed in *C. tetani* 454, we generated a pMTL expression construct encoding catalytically inactive BoNT/A1^ERY^ (pMTL82153-BoNT/A1^ERY^) and conjugated it into *C. tetani* 454 via the *E. coli* donor strain CA434.

*C. tetani* 454 (82153-A1^ERY^) was cultured alongside *C. botulinum* Hall A hyper/tox-(83152-A1^ERY^), which has previously been shown to produce high levels of rBoNT/A complex [41,56,57]. *C. botulinum* Hall A hyper/ tox-retains genes encoding BoNT/A NAPs allow for a comparison of BoNT/A1^ERY^ expression in *Clostridial* strains with and without native complexing proteins. Samples were cultured in TPGY, modified Mueller Miller media (+reduced iron) and toxin production media (TPM) for 24–96 h. Modified Mueller Miller media is commonly used to support TeNT production in *C. tetani*, TPM is commonly used to express BoNTs in *C. botulinum* strains and TPGY is a general clostridial growth media.

Western blot analysis demonstrated that fermentation in TPGY resulted in a small amount of 150 kDa full-length rBoNT/A1^ERY^ in both *C. tetani* 454 and *C. botulinum* Hall A hyper tox-, but a large number of smaller bands indicating degradation were observed, suggesting that TPGY may not be an optimal expression media. In *C. tetani* 454 (82153-A1^ERY^) cultured in TPM, full-length rBoNT/A1^ERY^ was detected as early as 24 h in the cell pellet with increasing amounts detected in whole culture samples by 144 h (Figure 2A). The toxin appeared to be partially post-translationally processed into the active dichain form at 144 h, as evidenced by the separation of the ~150 kDa holotoxin into ~100 kDa heavy chain and ~50 kDa light chain after reduction in the sample by DTT (Figure 2B). This demonstrates that a recombinant complex-free botulinum toxin can be expressed by a *C. tetani* strain and can be converted to its activated dichain form via endogenous proteases. However, compared to the production of the rBoNT/A1^ERY^ complex by *C. botulinum* Hall A hyper tox-, production in *C.tetani* grown in TPM was lower and occurred later during culture (Figure 2). In both cultures, a significant portion of the toxin appeared degraded as evidenced by smaller MW bands on the Western blot (Figure 2).

Similarly, when cultured in modified Mueller Miller (+reduced iron) media, the full-length toxin expressed by *C. tetani* 454 (82153-A1^ERY^) appeared more degraded than that of rBoNT/A1^ERY^ expressed in *C. botulinum* Hall A hyper/tox-(Figure 3A). Prior studies have reported that the mechanism by which reduced iron powder increases *C. tetani* growth and toxin production may be providing the presence of an insoluble surface area [53]. The examination of rBoNT/ A1^ERY^ expression by *C. tetani* 454 (82153-A1^ERY^) on modified Mueller Miller media agar plates without the addition of reduced iron resulted in strong toxin production with minimal break-down (Figure 3B), supporting the presence of an insoluble growth surface as playing a role in *C. tetani* growth and toxin production.

### 3.4. The Role of pH in CNT Expression and Stability

A previous study indicated that a culture pH below pH 6 plays a critical role in BoNT complex stability produced by *C. botulinum* [41]. In this study, the culture pH varied depending on growth conditions (Appendix A). *C. tetani* 454 (82153-TeNT) cultured in Mueller Miller (+reduced iron) had an average (n = 3) culture media pH of ~5.2 at 24 h and ~4.9 by 144 h with TeNT production detected throughout the time period (Figure 1). When the same sample was cultured in TPM, the pH was similar, averaging ~5 at 24 h and ~4.9 at 144 h. However, *C. tetani* 454 (82153-A1^ERY^) cultured in modified Mueller Miller (+reduced iron) had an average pH of ~6.7 at 24 h, which dropped to ~6.2 by 144 h, and when cultured in TPM, the samples maintained a pH of ~6.9–7.0 from 24–144 h. This trend was very similar for the negative, toxin-free expression control *C. tetani* 454 (82153), indicating that the *tent* gene cluster itself might contribute to the metabolically induced pH drop of endogenous *C. tetani* [35,54,55].

### 3.5. BoNT/A1 Holotoxin Is More Stable in Spent Modified Mueller Miller Media than in Spent TPM

In order to directly examine pH dependence of complex-free BoNT/A1 and TeNT stability in culture supernatants, the rCNTs were incubated in spent media of the negative expression control *C. tetani* 454 (82153) as well as *C. botulinum* LNT01 cultured in modified Mueller Miller (+reduced iron) or TPM for 4 days. Unlike *C. botulinum* Hall A hyper/tox-, *C. botulinum* LNT01 lacks the entire *bont* gene cluster including all complexing protein genes [33]. This allowed examination of the effects of culture pH on BoNT/A and TeNT without the possibility of any stabilizing effect from type-A complexing proteins. After 24 h incubation of BoNT/A1 or TeNT in sterile filtered spent culture media adjusted to pH 5, 6, 7, or 8, remaining 150 kDa BoNT/A1 and TeNT was assessed by Western blot (Appendix A) and densitometry (n = 3), and in all cases, the pH of spent media had little or no impact on CNT stability (Figure 4). However, a large difference was noticed between the two tested spent media formulations. The spent TPM of *C. tetani* 454 resulted in almost no detectable 150 kDa toxin but rather primarily smaller protein bands consistent with specific cleavage of the protein toxin (Appendix A). Less protein break-down and more 150 kDa protein was observed in spent TPM of *C. botulinum* LNT01; however, even in that case, the amount of 150 kDa TeNT or BoNT was reduced to 20–40% with distinct break-down products observed for BoNT/A1 but not for TeNT (Appendix A). Both BoNT/A1 and TeNT exhibited far less degradation in spent modified Mueller Miller media than in spent TPM with between 60 and 80% of the added protein amount remaining as the 150 kDa band (Figure 4), although few distinct break-down products were still clearly visible (Appendix A). Interestingly, some break-down of the 150 kDa BoNT/A1 and TeNT even in fresh media could not be ruled out, although observed decreases in 150 kDa bands were much lower in fresh media than in the spent media, and almost no break-down products were observed on the Western blot (Appendix A). Instead, weak larger bands were observed for both toxins in fresh media, indicating the potential that minor degradation, auto-proteolysis, or aggregation may be due to media components present in both TPM and Mueller Miller media.

## 4. Discussion

Of all the clostridial neurotoxins, TeNT is unique in that it is not encoded in a gene cluster alongside non-toxic neurotoxin associated proteins (NAPs) and does not require NAPs to be produced as a stable protein in fermentation cultures [1]. Free BoNTs (without NAPs), on the other hand, are readily degraded by environmental and endogenous proteases [21,58], whereas complexing with NAPs to form large, multi-protein complexes prevents protein break-down under certain conditions and increases BoNT oral toxicity [28,35,59]. The production of TeNT by *C. tetani* is tightly controlled by a complex network of regulatory genes [35,54,55] and has been optimized for the large-scale production of stable and soluble protein to be processed for the widely used inactivated tetanus protein vaccine [42,60]. The most commonly used production media for TeNT is modified Mueller Miller [34].

Given the structural and functional similarities between BoNT and TeNT holotoxins, we hypothesized that the intra and extracellular conditions of *C. tetani* may be responsible for toxin stabilization and would support the recombinant expression of BoNTs without their native complexing proteins. To our knowledge, our data demonstrate the first recombinant expression of a full-length and proteolytically activated rBoNT by a *C. tetani* strain.

We investigated the expression of rTeNT and rBoNT/A1^ERY^ production in *C. tetani* using the modular pMTL8000 expression vector system [47] combined with two *C. tetani* strains that are closely related to the Harvard strain [44] used for the large-scale production of TeNT [60]. Promotors for thiolase (*thl*; resulting in pMTL82152) and ferredoxin (*fdx*; resulting in pMTL82153), which are often used for high-level gene expression in *Clostridial* species [55], as well as a native positive regulator of *tent*, *tetR* (resulting in pMTL82155), were examined [30,31]. Despite successful cloning and plasmid sequence verification, certain pMTL expression constructs appeared to be incompatible for transformation into *E. coli* donor strains. Specifically, pMTL82152-TeNT, pMTL 82152-A1^ERY^, and pMTL82155-A1^ERY^ displayed greatly increased rates of plasmid rearrangement following transformation into conjugative *E. coli* donor strains CA434 or S. Express. Whole plasmid sequencing of the plasmids isolated from *E. coli* revealed full or partial loss of the toxin gene and maintenance of the antibiotic resistance gene. In some cases, this gene loss and rearrangement occurred less than five hours post-transformation. Ingle et al. [61] discuss similar instances of plasmid instability of plasmids containing genes under *fdx* and *thl* promoters and propose that the high expression of certain genes downstream of strong constitutive promotors is likely detrimental to *E. coli*. The repeated loss of r*cnt* genes from plasmids pMTL82152-TeNT, pMTL82152-A1^ERY^, and pMTL82155-A1^ERY^ after transformation into *E. coli* donor strains indicates potential selection pressure against these toxin genes in *E. coli*.

*C. tetani* 454/19406 (82153-TeNT) expressed rTeNT at higher quantities and with fewer break-down products than either the wild-type strains or *C. tetani* 454 (82155-TeNT), which contains the endogenous *tent* promotor and regulator region, *tetR* (Figure 1, Appendix A). In addition, no metabolic shift [35,54,55,62] was required for the high-level production of rTeNT under the fdx promoter. This was the case in all three culture media—modified Mueller Miller (+reduced iron) [42], HySoy, and HySoy-T—although modified Mueller Miller media consistently resulted in the highest expression levels (Figure 1). A previous study found that the addition of reduced iron powder to modified Mueller Miller culture media resulted in ~6 times greater production of TeNT compared to media lacking reduced iron [53].

Interestingly, native TeNT expressed by wild-type strains and rTeNT expressed by *C. tetani* under the control of the endogenous *tent* promoter region including *tetR* (82155-TeNT) was largely undetected in whole culture until ~72 and 96 h, coinciding with a culture pH drop from neutral to ~4.8 (Appendix A). In *C. tetani* Harvard strain E88, Licona et al. [62] reported that *tetR* and *tetX* expression is extremely low until the cell shifts from consuming free amino acids to peptides between 10 and 40 h, and that the upregulation in toxin synthesis genes corresponded with an increase in culture pH to ~7.2 [62]. Other studies also observed an increase in culture pH prior to cell lysis and the release of TeNT into the supernatant [63,64]. It is unclear whether the pH decrease we observed is due to growth in static cultures, as opposed to an agitated bioreactor, or whether this is a product of the recombinant protein expression. *C. tetani* 454/19406 (82153-TeNT) samples, which expressed *tent* under the control of the ferrodoxin promoter, contained high levels of rTeNT in the whole culture by 24 h with a culture pH of 4.9–5.3 maintained throughout the duration of the expression studies (Appendix A). The temporary difference in both culture pH and TeNT production between *C. tetani* 454 (82155-TeNT) and 454 (82153-TeNT), where the only difference is the promotor region, indicates that the *tetR* promotor/regulator region or the expressed TeNT itself may contribute to the metabolism of *C. tetani*, prompting a pH shift to acidic levels at the same time that *tent* is upregulated [55]. This hypothesis is consistent with *C. tetani* 454 (82155-TeNT) and the wild-type strain WR23 both displaying similar TeNT expression timelines and pH changes to culture media and the *C. tetani* 454 (82153) control strain not displaying any pH shift (Appendix A), and it raises the interesting question of a more global metabolic role for the TeNT or, more likely, Tet^R^, beyond the regulation of TeNT production.

In order to examine the production of NAP-free rBoNT in the *C. tetani* expression system, the catalytically inactive *bont/a1^ERY^* was cloned into the pMTL82153 under control of the ferredoxin promoter (pMTL82153-A1^ERY^), and the rBoNT/A1^ERY^ expression in *C. tetani* 454 (82153-A1^ERY^) was analyzed in comparison to the production of the BoNT/A1^ERY^ complex in *C. botulinum* Hall A hyper/tox- (83152-A1^ERY^or both *C. tetani* and *C. botulinum*). NAP-free BoNT/A1^ERY^ was produced in *C. tetani* 454 at about 50% reduced (TPM) or similar (MM + reduced iron) levels as the BoNT/A1^ERY^ complex in *C. botulinum* Hall A hyper tox with evidence of at least partial proteolytic nicking to the dichain form that increased with increased culture time (Figure 2 and Figure 3A). However, an increased amount of smaller protein bands, indicating higher rates of protein degradation, were observed for the NAP-free BoNT/A1^ERY^ produced in *C. tetani* 454 than for the BoNT/A1^ERY^ complex produced in *C. botulinum* Hall A hyper/tox- in both media. The break-down products for BoNT/A1^ERY^ produced in *C. tetani* vs. *C. botulinum* varied, indicating either the presence of distinct proteases in the spent media of the two organisms or the availability of distinct protease cleavage sites in complexed versus free BoNT/A1. Direct examination of the stability of BoNT/A1 and TeNT holotoxins (150 kDa, complex free) in spent culture media from a *C. botulinum* strain LNT01, which does not express any complexing proteins, and *C. tetani* 454 indicated similar stability in spent media from both strains but a strong media type dependence (Figure 4). Both toxins showed only mild partial degradation in spent Mueller Miller but strong degradation in spent TPM (Figure 4, Appendix A). No significant pH dependence for degradation was observed under any condition. This is in contrast to a previous study that showed strong pH-dependent proteolytic degradation of the BoNT/A1 complex in spent media of *C. botulinum* strain Hall A hyper [41], emphasizing the protective role of NAPs and the difference in conditions required for the production of complex free versus complexed rBoNTs.

The unique pattern of degradation bands observed for both BoNT and TeNT in spent TPM and Mueller Miller media (Appendix A) indicates the differential production or secretion of rCNT degrading proteases induced by the growth media in both *C. tetani* and *C. botulinum* and emphasizes the role of nutrient-induced bacterial metabolism on toxin production by affecting toxin stability post-translation. The unexpected observation of fewer break-down bands for both rTeNT and rBoNT/A1^ERY^ in spent MM media of *C. botulinum* LNT01 than *C. tetani* 454 (Appendix A) suggests that either *C. botulinum* strain LNT01 produces less CNT specific protease activity or that degradation is more complete to where fewer bands are detected on the gel. The fact that there was no significant difference in the amounts of remaining 150 kDa rCNT bands (Figure 4) supports the latter. While these studies indicate that complex-free TeNT and BoNT stability is largely independent of pH (Figure 4), a correlation of acidic pH and greater toxin production was observed in culture (Figure 2 and Figure 3), indicating that metabolic changes associated with pH changes may result in the production of specific sets of proteases.

Overall, the relatively good stability of 150 kDa rBoNT/A1^ERY^ in spent Mueller Miller media of either *C. tetani* or *C. botulinum* (on average 60–80%) (Figure 3A) also suggests that *C. tetani* strain 454 as well as other atoxic *clostridial* strains grown in specific optimized media may be utilized as a new expression system for the production of complex-free recombinant CNTs for functional analysis or vaccine development. As such, an expression system is not dependent on specific NAPs for proteins of the rBoNT; the system will likely be applicable for any BoNT sero and subtypes.

Some BoNTs are not easily purified from their native *C. botulinum* strains in quantities sufficient for structural and functional analysis due to low expression levels or due to being produced by a dual toxin-producing strain. In addition, mutational studies require recombinant BoNT production. However, recombinant BoNT expression in an endogenous expression host is an invaluable, but underdeveloped, tool for the further functional characterization of BoNTs. Current methods for rBoNT expression rely either on *E. coli*, which lacks the ability to proteolytically process and activate the toxin [65], or atoxic *C. botulinum* type A strains, which retain native type A NAPs and have only successfully produced BoNT/A serotypes [57]. In a recent study, we found significant differences in the potency of rBoNTs produced in heterologous versus endogenous expression systems [66], although the reasons underlying these potency differences are currently unknown. This study adds a novel recombinant expression system for rBoNTs utilizing an atoxic *C. tetani* or *C. botulinum* strain.

In summary, this study demonstrated that the stable production of NAP-free TeNT by *C. tetani* is primarily controlled by expression strain properties and metabolic pathways that can be regulated by media formulation, and that NAP-free CNT stability in growth media is independent of pH. This study also indicates that atoxic *C. tetani* or other atoxic clostridia can be utilized as a novel clostridial expression system for the production of NAP-free rBoNTs and TeNT. The initial optimizations performed in this study to select the expression vector, promotor, atoxic *C. tetani* strain, and media formulations for the highest production levels of rCNTs lay the groundwork for this novel expression system, and additional strain-specific media and growth condition optimizations have the potential to further improve this system as a universal NAP-free CNT expression system.

## Figures and Tables

**Figure 1 microorganisms-12-02611-f001:**
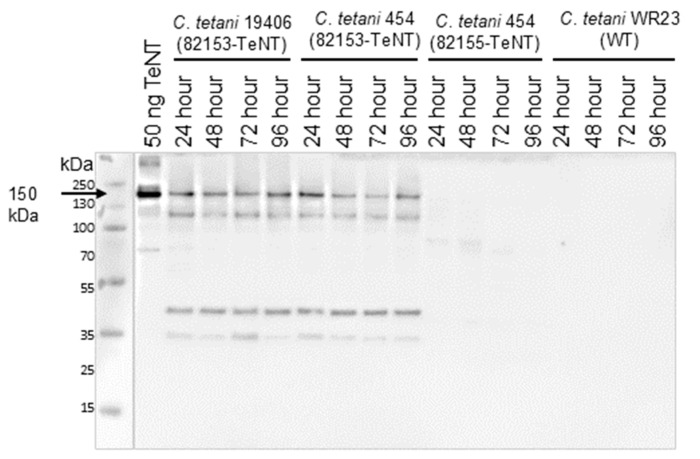
*C. tetani* 454 and 19406 express stable, full-length rTeNT at levels exceeding that of a wild-type strain under the ferredoxin promoter.

**Figure 2 microorganisms-12-02611-f002:**
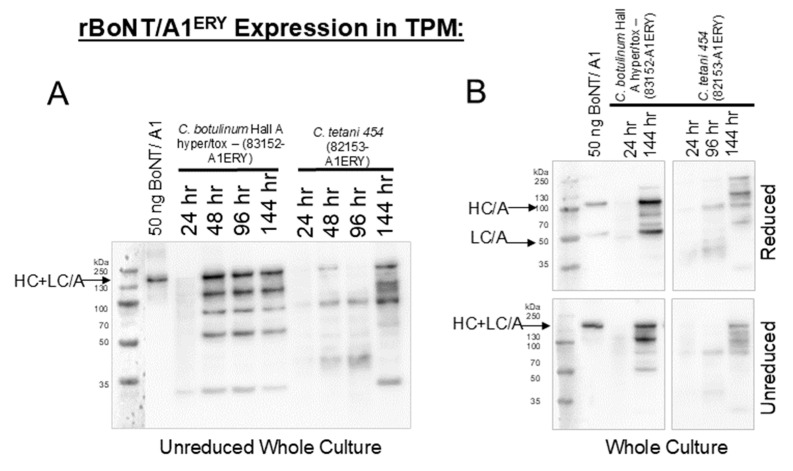
Full-length rBoNT/A1^ERY^ is expressed by *C. tetani* 454 but is only partially proteolytically processed to its dichain form. (**A**) Western blots depict the stability of rBoNT/A^ERY^ over time when the expression strains are cultured in toxin production media (TPM). *C. botulinum* Hall A Hyper/tox-(82152-A1^ERY^) is known to strongly express stable BoNT/A and was therefore used as an expression control to compare against *C. tetani* 454 (82153-A1^ERY^). (**B**) Whole culture samples were collected at the noted times and either reduced with 100 mM DTT or left unreduced. The 50 ng purified BoNT/A1 was used as a loading control. The locations of the ~150 kDa holotoxin (HC + LC/A), the ~100 kDa heavy chain (HC/A), and the ~50 kDa light chain (LC/A) are indicated.

**Figure 3 microorganisms-12-02611-f003:**
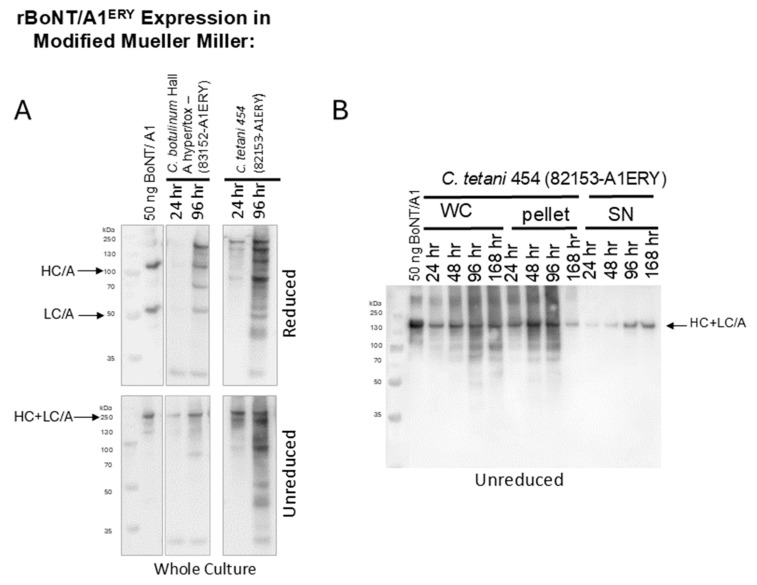
Full-length rBoNT/A1^ERY^ is expressed by *C. tetani* 454 in modified Mueller Miller media but degrades over time. (**A**) Western blots of *C. botulinum* Hall A hyper/tox- (83152-A1^ERY^) and *C. tetani* 454 (82153-A1^ERY^) cultured in modified Mueller Miller media (+0.5 g/L reduced iron powder). Whole culture samples were taken at 24 and 96 h and either reduced with 100 mM DTT or left unreduced. (**B**) Western blot of *C. tetani* 454 (82153-A1^ERY^) expression on modified Mueller Miller agar plates without the addition of reduced iron powder or any alternative iron source. Whole cell growth (WC) on the plate was resuspended at the indicated time points and centrifuged into cell pellet (pellet) and supernatant (SN). Then, 50 ng purified BoNT/A1 was loaded as a control. The locations of the ~150 kDa holotoxin (HC + LC/A), the ~100 kDa heavy chain (HC/A), and the ~50 kDa light chain (LC/A) are indicated.

**Figure 4 microorganisms-12-02611-f004:**
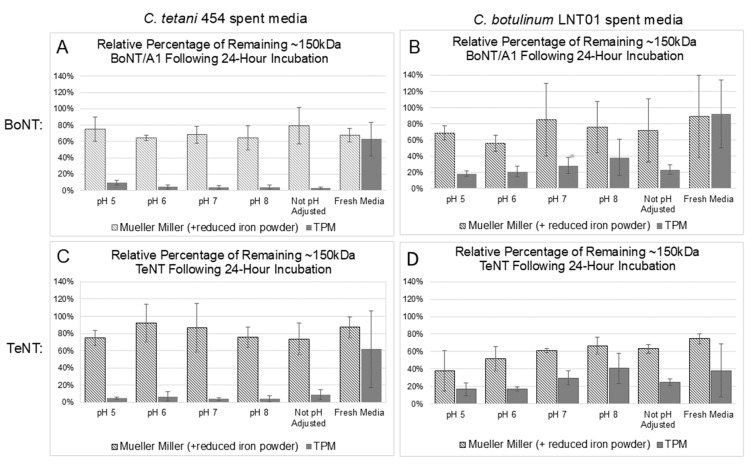
Comparative stability of clostridial neurotoxins in spent media under varying pH conditions. Static cultures of the atoxic strains *C. tetani* 454 (**A**,**C**) and *C. botulinum* LNT01 (**B**,**D**) were incubated at 37 °C for 96 h in either modified Mueller Miller media (supplemented with 0.5 g/L of reduced iron powder) or toxin production media (TPM). Culture supernatant was collected and pH adjusted to pH 5, 6, 7, and 8 via 1 M NaOH or 1 M HCl. The pH adjusted spent media, spent media with no pH adjustment, and media which had not supported bacterial growth (fresh media) were sterile filtered through a 0.22 uM PVDF membrane. Purified 150 kDa BoNT/A1 (**A**,**B**) or TeNT (**C**,**D**) were added to each sample, incubated for 24 h at 37 °C, and then analyzed via SDS-PAGE and Western blot. The ~150 kDa protein bands, indicating the amount of full-length toxin remaining, were analyzed via densitometry. The mean concentration (n = 3) of remaining holotoxin is represented as a percentage relative to the amount of toxin initially added to the sample.

## Data Availability

The original contributions presented in this study are included in the article/Appendix A. Further inquiries can be directed to the corresponding author(s).

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
