# Peer review of "Expression of Recombinant Clostridial Neurotoxin by C. tetani"

_microorganisms, 2024, doi:10.3390/microorganisms12122611_

Round 1
Reviewer 1 Report
Comments and Suggestions for Authors
Dear authors,
The paper is very well presented and written. It covers an original topic which has not been well explored and can be very important in the field.
Please find comments and suggestions for the authors:
Abstract:
Important to mention also that the BoNTs can be produced in proteolytic form or non-proteolytic form depending on the dichain or monochain form.
Line 12: the complex-free structure of TeNT may be in line with its in vivo fate: intra-neuronal and not submitted to gastrointestinal enzymes.
Line 21: "found that the stability"
Line 40: please also mention the Tetanus neonatorum (most frequent cause) due to poor umbilical hygiene.
Line 52: please mention the nerve terminal types targeted by TeNT as opposed to the cholinergic types for BoNT.
Line 61: and also C. Baratii.
Please also mention that BoNT genes were recently found in non-Clostridial species thanks to bioinformatics tools.
Line 92: would be interesting to know if the TeNT production levels are significantly higher than those of BoNT in culture. This could partly explain the differences.
Methods:
Line 118: C. Botulinum in italic.
Line 148: the peptone levels and types are critical in the production (unpublished results). Would be interesting to know if peptone levels were comparable in the culture media for TeNT or BoNT productions.
Line 222: what was the rough percentage of unstable plasmids?
Line 252: 95°C
Line 270: stability maybe assessed at longer incubations times since production of BoNT occurs within days.
Results:
Lines 317-326: a table showing the composition of each media (Mueller, Latham and Demain) would be interesting for the reader to find out if some component influence the toxin production.
Line 398: was the pH similar in C.tetani culture to the pH in C.Bot culture? pH may influence this degradation level?
Line 445: what components in TPM media could influence de degradation in comparison to the Mueller medium?
Discussion:
Line 177: "to be processed for the formulation of the widely used tetanus protein vaccine"
Line 565: indicates
Interesting to add in this paragraph suggestions about the role of some media components.
Also, it would be beneficial to discuss the benefit of this new production model for the production of BoNT vaccines candidates based on atoxic rBoNTs.
Author Response
Response to Reviewer Comments:
We’d like to thank the reviewers for their careful and thorough review of the manuscript. We have addressed the comments by making changes in the manuscript and have detailed our response below:
Reviewer 1:
Important to mention also that the BoNTs can be produced in proteolytic form or non-proteolytic form depending on the dichain or monochain form.
We agree, and this is now addressed in lines 63-65.
Line 12: the complex-free structure of TeNT may be in line with its in vivo fate: intra-neuronal and not submitted to gastrointestinal enzymes.
Yes, this may be the case. We do discuss the role of NAPs in protecting BoNT (Line 77-78) as it passes through the GI tract, whereas TeNT enters the body through wounds and is not exposed to that harsh environment.
Line 21: "found that the stability"
Thank you, this has been changed.
Line 40: please also mention the Tetanus neonatorum (most frequent cause) due to poor umbilical hygiene.
Thank you for this excellent suggestion. We have added in a mention of neonatal tetanus, and how the adoption of the tetanus vaccine has reduced adult and neonatal case numbers (line 56).
Line 52: please mention the nerve terminal types targeted by TeNT as opposed to the cholinergic types for BoNT.
Thank you, we’ve mentioned in line 47 that TeNT targets the inhibitory interneurons in the CNS.
Line 61: and also C. Baratii.
- baratii has been added.
Please also mention that BoNT genes were recently found in non-Clostridial species thanks to bioinformatics tools.
We have now mentioned that bioinformatic databases aided in the discovery of many novel bont genes (Line 72).
Line 92: would be interesting to know if the TeNT production levels are significantly higher than those of BoNT in culture. This could partly explain the differences.
The quantity of TeNT production differs based culture conditions, and literature indicates peak production levels similar to those of BoNT. On our Western blots, we have included known quantities of purified toxin to compare with the toxin expressed in C. tetani or C. botulinum. Based on this, it also does not appear the WT or rTeNT is expressed at a much greater level than rBoNT is expressed by C. botulinum.
Line 118: C. Botulinum in italic.
This change has been made
Line 148: the peptone levels and types are critical in the production (unpublished results). Would be interesting to know if peptone levels were comparable in the culture media for TeNT or BoNT productions.
In this study, the exact same media formulations were used for expression of rBoNT and rTeNT. The method section was edited to clarify this.
Line 222: what was the rough percentage of unstable plasmids?
The instability of the plasmid greatly varied from construct to construct. On average, we screened 12 putative transformants and would get between 0-10 clones containing the intact pMTL plasmid construct. In some cases, a shorter incubation time following transformation into the donor E. coli strain resulted in more intact plasmids. However, we did not conduct detailed studies determining percentage of unstable plasmids and thus do not feel comfortable providing a number.
Line 252: 95°C
This change has been made.
Line 270: stability maybe assessed at longer incubations times since production of BoNT occurs within days.
Thank you, this would indeed be an interesting experiment to perform in the future. In previous studies, we did not observe an additional decrease in BoNT stability in spent C. botulinum media after 24 h.
Results:
Lines 317-326: a table showing the composition of each media (Mueller, Latham and Demain) would be interesting for the reader to find out if some component influence the toxin production.
We have added the media composition for the modified Mueller Miller media to the materials section (Line 155) to make comparison easier for the readers.
Line 398: was the pH similar in C.tetani culture to the pH in C.Bot culture? pH may influence this degradation level?
This is a very interesting point that we did examine. The pH values differed depending on the expression host organism as well as the growth media. In order to investigate this more closely, we performed the toxin stability experiment across a range of pH values (Figure 4, Supplemental Figure 3), which demonstrated similar toxin stability across the pH tested. It may be the case though, that culture pH differences also prompt differential expression of the toxin and/or proteases targeting the toxin, and certainly differences in culture pH indicate differences in metabolism. This topic is discussed in section 3.4 and table 2.
Line 445: what components in TPM media could influence de degradation in comparison to the Mueller medium?
We have not yet investigated the role of individual media components on Clostridial metabolism or toxin expression/stability. However, there was very little degradation in fresh media, thus media components might only have a minor contribution to instability compared to metabolites or a specific environment produced by the bacteria during growth in the specific media. This paragraph was slightly updated to clarify.
Discussion:
Line 177: "to be processed for the formulation of the widely used tetanus protein vaccine"
This change has been made.
Line 565: indicates
This change has been made.
Interesting to add in this paragraph suggestions about the role of some media components.
Please see our answer for line 445
Also, it would be beneficial to discuss the benefit of this new production model for the production of BoNT vaccines candidates based on atoxic rBoNTs.
Thank you for the suggestion, we have now mentioned that atoxic Clostridial strains could be utilized for expression of atoxic rCNTs as vaccine candidates (line 582).
Reviewer 2 Report
Comments and Suggestions for Authors
In this manuscript, Brieana et al. address the intriguing question of why TeNT and BoNTs are both neurotoxins but exhibit markedly different behavior in Clostridium tetani. They have developed and optimized strategies for the recombinant expression of full-length and proteolytically activated rBoNT using a C. tetani strain. This is an interesting study that provides valuable strategies for advancing C. tetani clostridial neurotoxin research. I recommend the paper for publication after addressing the following major comments:
- Original Image Files for Figures
In Fig. 1, there appears to be a grey lane between the marker and the sample lanes. Is this the original image for the Western blot or a merged version? Please ensure that the original, unmodified image is provided. A similar issue occurs in Fig. 2A, where it looks like the marker has been merged with other lanes. - Expression and Functional Validation of Mutants
Could the authors consider expressing mutants of these two proteins to verify their expression levels and functional behavior? For instance, a mutant targeting the cysteine residue responsible for the disulfide bond formation between LC and HC would be particularly insightful. While not essential, this addition would strengthen the study.
Minor comments:
1. Please check correct description of the background. For example, Line 42-43, as you mentioned the domain information about LRRK2LRRK2 have multiple domains, more than one kinase domain et.al. your description is not right. Please make a correction description about it.
2. Please check the writing error throughout the paper, for example, line 80 “ in interaction” should be “interaction”, right? The red line is highlighted in the figure 1A. please carefully check and figure out all the writing errors.
Author Response
Response to Reviewer Comments:
We’d like to thank the reviewers for their careful and thorough review of the manuscript. We have addressed the comments by making changes in the manuscript and have detailed our response below:
Reviewer 2:
Original Image Files for Figures
In Fig. 1, there appears to be a grey lane between the marker and the sample lanes. Is this the original image for the Western blot or a merged version? Please ensure that the original, unmodified image is provided. A similar issue occurs in Fig. 2A, where it looks like the marker has been merged with other lanes.
Thank you, we have now supplied the original Western blot images. The grey line between the marker and samples is indeed due to fusion of two images. This is done automatically in the Azure imager, which images the same membrane using chemiluminescence to see the sample bands and visual light to image the Marker bands. The two images are then fused by the imager software to produce the original unedited image of the entire Western blot with protein size marker.
Expression and Functional Validation of Mutants
Could the authors consider expressing mutants of these two proteins to verify their expression levels and functional behavior? For instance, a mutant targeting the cysteine residue responsible for the disulfide bond formation between LC and HC would be particularly insightful. While not essential, this addition would strengthen the study.
We thank you for your suggestions and we agree this will be a very interesting future project. We do feel that it falls outside the scope of this current manuscript.
Comments on the Quality of English Language
Minor comments:
Please check correct description of the background. For example, Line 42-43, as you mentioned the domain information about LRRK2LRRK2 have multiple domains, more than one kinase domain et.al. your description is not right. Please make a correction description about it.
Please check the writing error throughout the paper, for example, line 80 “ in interaction” should be “interaction”, right? The red line is highlighted in the figure 1A. please carefully check and figure out all the writing errors.
These comments seem to not apply to our manuscript and we assume they have been submitted by error.